# Crop Wild Relatives Crosses: Multi-Location Assessment in Durum Wheat, Barley, and Lentil

Noureddine El Haddad [1,2], Miguel Sanchez-Garcia [1], Andrea Visioni [1], Abderrazek Jilal [3], Rola El Amil [4], Amadou Tidiane Sall [5], Wasihun Lagesse [6], Shiv Kumar [1] and Filippo M. Bassi [1,*]

1   International Center for Agricultural Research in the Dry Areas (ICARDA), Rabat 10112, Morocco; n.el-haddad@cgiar.org (N.E.H.); m.sanchez-garcia@cgiar.org (M.S.-G.); a.visioni@cgiar.org (A.V.); sk.agrawal@cgiar.org (S.K.)
2   Faculté des Sciences, Université Mohammed V in Rabat, Rabat 10112, Morocco
3   National Institute of Agricultural Research (INRA), Rabat 10112, Morocco; abderrazek.jilal@inra.ma
4   Lebanese Agricultural Research Institute (LARI), Zahle 287, Lebanon; ramil@lari.gov.lb
5   Institut Sénégalais de Recherches Agricoles (ISRA), Saint-Louis 46024, Senegal; tidianesall11@yahoo.com
6   Ethiopian Institute Agricultural Research (EIAR), Addis Ababa 1000, Ethiopia; wasihunl@yahoo.com
*   Correspondence: f.bassi@cgiar.org

**Abstract:** Crop wild relatives (CWR) are a good source of useful alleles for climate change adaptation. Here, 19 durum wheat, 24 barley, and 24 lentil elites incorporating CWR in their pedigrees were yield tested against commercial checks across 19 environments located in Morocco, Ethiopia, Lebanon, and Senegal. For each crop, the combined analysis of variance showed that genotype (G), environment (E), and genotype x environment (G×E) effects were significant for most of the traits. A selection index combining yield potential (G) and yield stability (G×E) was used to identify six CWR-derived elites for each crop matching or superior to the best check. A regression analysis using a climate matrix revealed that grain yield was mostly influenced by the maximum daily temperature and soil moisture level during the growing stages. These climatic factors were used to define five clusters (i.e., E1 to E5) of mega-environments. The CWR-derived elites significantly outperformed the checks in E1, E2, and E4 for durum wheat, and in E2 for both barley and lentil. The germplasm was also assessed for several food transformation characteristics. For durum wheat, one accession (Zeina) originating from *T. araraticum* was significantly superior in mixograph score to the best check, and three accessions originating from *T. araraticum* and *T. urartu* were superior for Zn concentration. For barley, 21 accessions originating from *H. spontaneum* were superior to the checks for protein content, six for Zn content, and eight for β-glucan. For lentil, ten accessions originating from *Lens orientalis* were superior to the check for protein content, five for Zn, and ten for Fe concentration. Hence, the results presented here strongly support the use of CWR in breeding programs of these three dryland crops, both for adaptation to climatic stresses and for value addition for food transformation.

**Keywords:** yield stability; crop wild relatives; durum wheat; barley; lentil; nutritional quality; genotype x environment interaction; drought stress; heat stress

## 1. Introduction

Durum wheat (*Triticum durum* Desf.), barley (*Hordeum vulgare* L.), and lentil (*Lens culinaris* Medik. *culinaris*) are three important food crops. Their importance is even greater in the dryland of many developing countries, where they represent true staples for human and livestock nutrition. These crops provide significant amounts of the daily intake of calories, protein, and rich sources of micronutrients such as zinc (Zn) and iron (Fe) [1–3]. However, the rise of global temperatures and the significant reduction in annual precipitation are threatening the cultivation of these crops, especially in the drylands [4–6].

The impact of heat stress and/or water stress have been investigated in durum wheat [7–9], barley [10,11], and lentil [12–14]. Both stresses obstruct an array of processes

including growth, floral development, carbohydrate reallocation, protein concentration in the grains, lipids, and micronutrient (zinc and iron) content, which ultimately affects grain and straw yield and quality [14–16]. The reproductive stage has been identified as the most sensitive stage to these stresses, which can lead to a dramatic reduction in seed numbers by affecting pollen and ovule function [17–19]. Therefore, breeders have set off to develop new varieties better adapted to these climatic constraints, while at the same time trying to improve nutritional and transformation characteristics to add value to the harvest [20].

Crop wild relatives (CWR) are undomesticated species related to the cultivated crops that are considered to be a rich source of untapped genetic diversity that can be exploited to favor the climate adaptation of crops [21–25]. In fact, the "wild" nature of these species obliges them to survive without the support of farmers, facing several climatic constraints in marginal or hostile soils [2,26,27]. It is estimated that 30–50% of the favorable alleles in modern breeding lines were contributed by wild relatives [28,29].

Durum originated approximately 12,000 years ago in the Fertile Crescent, when ancient farmers selected among cultivated forms of wild emmer *Triticum turgidum* ssp. *diccocoides*, which is itself derived from a naked type that was easier to thresh (*Triticum turgidum* ssp. *dicoccum*) [30–33]. Several negative phenotypic variations occurred during domestication, including a loss of early vigor, a reduction in the number of tillers, and a change in root–shoot ratio [34,35]. Barley also originated from the domestication of its wild form *H. spontaneum* and today, modern barley still contains about 40% of the original wild alleles [36]. The domestication process also caused the loss of several useful traits by affecting major genes of resistance to abiotic and biotic stresses [37]. Sharma et al. [38] investigated the genetic diversity of 25 diverse exotic genotypes against cultivated barley to identify several beneficial alleles for grain parameters, especially grain length, showing the potential to improve this trait. Lentil is regarded as a founder crop of the Neolithic Near Eastern agriculture [39,40]. *Lens culinaris* ssp. *orientalis* is considered to be the wild progenitor of the cultivated lentil [2,41,42]. The domestication of lentil led to a loss of genetic diversity of approximately 40% [43], which explains the increased susceptibility of the cultivated varieties to biotic and abiotic stresses [44,45]. Martín-Robles et al. [46] reported that domestication reduced mycorrhizal responsiveness, influencing root morphological traits (such as root length and root diameter) and therefore nutrient uptake ability.

The usefulness of widening the genetic basis of durum wheat with CWR was demonstrated by Zaim et al. [25], who showed superior yield and quality characteristics of entries derived from *T. dicoccoides*, *Ae. speltoides*, and *T. araraticum*. Sall et al. [8] also reported the usefulness of wild emmer *T. dicoccoides* in improving the heat tolerance of modern germplasm. Finally, El Haddad et al. [47] demonstrated the superiority of CWR-derived elites in adapting to moisture stress and reduced fertilization. In barley, the use of *H. spontaneum* was shown to improve the response to major biotic and abiotic stresses [48–50]. Furthermore, higher concentrations of β-glucan, Zn, and Fe were identified in *H. spontaneum* derived lines [51–54]. Similarly, *L. orientalis* has been identified as an ideal source of superior alleles for heat and drought tolerance, as well as for higher protein and micronutrients concentrations [55–57].

In this study, CWR-derived elites of the three crops (durum wheat, barley, and lentil) have been compared individually against the most meaningful checks across locations to assess their adaptability response and the possible yield gain using wide-cross lines, and to evaluate the most important quality characteristics including protein and micronutrient content. The results presented here represent an in-depth assessment of the usefulness of CWR as breeding tool to help guide the next decade of genetic improvement, with a special focus on climatic adaptation and nutritional value.

## 2. Materials and Methods

### 2.1. Germplasm

Germplasm of durum wheat, barley, and lentil was assessed; the full list can be consulted in Table S1. The germplasm was from the International Center for Agricultural

Research in the Dry Area (ICARDA)'s own crosses, made to deliver usable genetic diversity to breeding programs. An indication of the hypothetical CWR contribution was calculated on the basis of the pedigree, assuming that in each round of crosses each parent contributes 50% of the genome of the progeny. For durum wheat, 19 CWR-derived elite lines were obtained from interspecific crosses with *T. aegilopiodes, Ae. speltoides*, *T. araraticum*, *T. dicoccoides*, and *T. urartu*. These were compared against five checks (Karim, Bonidur, Zagharin2, Chicca, and Kundermiki), which did not include any CWR. These checks represent released varieties or candidate varieties undergoing the final steps before release in Morocco, Lebanon, Ethiopia, and Senegal, and were selected to be representative checks across the test sites. For barley, 24 CWR-derived lines from *H. spontaneum* were used to challenge four commercial checks, two from Morocco (Rihane03 and Tamellalt), and one each from Australia (WI2291) and Syria (Furat-3). Chams was used as a local check in Lebanon stations, while Arsi was used in Ethiopia against six top-yielding CWR elites of barley. For lentil, 24 advanced lines emanating from crosses with *L. orientalis* were compared to the check Bakria (ILL 4605), which the most cultivated lentil cultivar in Morocco and has also been adopted in many Western and South Asia countries.

### 2.2. Study Sites and Management

The germplasm was assessed across 13 research stations located in Morocco, Lebanon, Ethiopia, and Senegal (Figure 1), over two successive crop seasons 2018–2019 (19) and 2019–2020 (20), for a total of 19 environments available for the test.

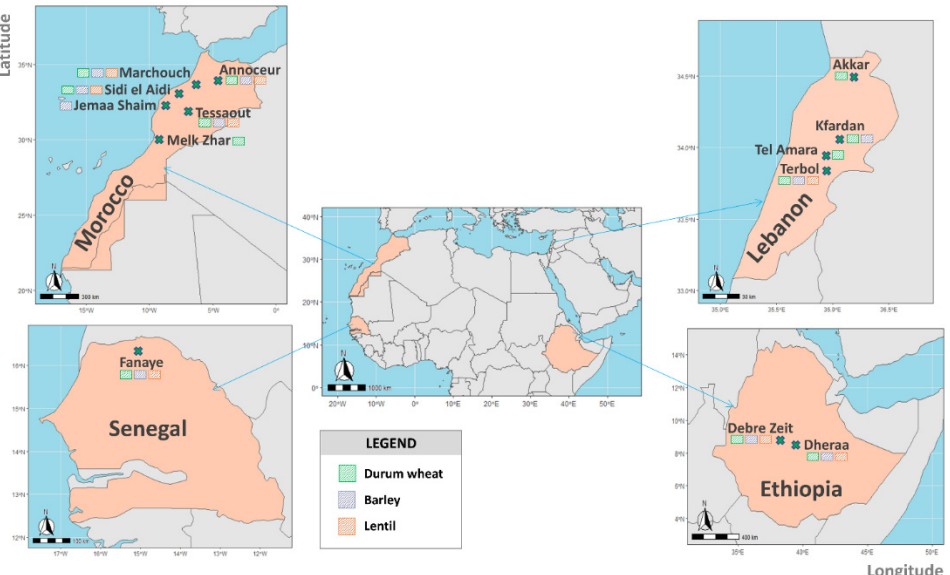

**Figure 1.** Geographical distribution of experimental farms used in this study. Square symbols are used to indicate the tested crop at each site.

In Morocco, six research farms were used in this study including the three drought-prone sites of Marchouch, Sidi el Aidi, and Jemaa Shaim, the two irrigated sites of Tessaout and Melk Zhar, and one site in the cold Atlas Mountains (Annoceur). A total of four locations were used in Lebanon: two with available supplemental irrigation (Terbol and Tel Amara), one drought-prone (Kfardan), and one high rainfall (Akkar). In Ethiopia, two sites were used, one with supplemental irrigation (Debre Zeit) and another typical of the drought-affected lowlands (Dheraa). In Senegal, one irrigated site was used (Fanaye), where maximum daily temperatures remain above 32 °C from planting to harvest. The main agroecological information for each station is reported in Table S2.

In the first season, durum wheat trials were planted at seven locations (Marchouch19, Sidi el Aidi19, Tessaout19, Annoceur19, Terbol19, Kfardan19, and Tel Amara19). Barley field experiments were planted at five experimental stations in Morocco (Marchouch19,

Sidi el Aidi19, Tessaout19, Annoceur19, and Jemaa Shaim19). In addition, one lentil trial was planted at Marchouch19. In the second season, the same set of durum wheat was repeated at five stations in Morocco (Marchouch20, Sidi el Aidi20, Tessaout20, Annoceur20, and Melk Zhar20), four in Lebanon (Tel Amara20, Kfardan20, Akkar20, and Terbol20), two in Ethiopia (Debre Zeit20 and Dheraa20), and one at the heat prone station in Senegal (Fanaye20). Likewise, the same set of 24 CWR-derived elites of lentil with Bakria as check was planted in Marchouch20, Sidi el Aidi20, Tessaout20, Annoceur20, Terbol20, Debre Zeit20, Dheraa20, and Fanaye20. The 28 lines of barley were planted at Marchouch20, Sidi el Aidi20, Tessaout20, and Annoceur20 in Morocco; however, only six top yielding CWR-derived lines were tested against two checks (Furat-3 and local check) at Terbol20, Kfardan20, Fanaye20, Debre Zeit20 and Dheraa20. In total, durum wheat, barley and lentil experiments were conducted independently at 19, 14, and 9 environments, respectively.

All trials consisted of plots of 7 m² planted in an alpha lattice design with two replications, except for the Marchouch19 for lentil, where four replications were used, with six, four and five incomplete blocks for durum wheat, barley, and lentil, respectively. All the trials were timely sown between 15 November and 15 December, with the exception of Ethiopia, where planting started in July to match the rainy season. Each crop was treated separately according to standard local practices. Details of field management of each crop at the test sites are presented in Table S3.

### 2.3. Data Recording

Grain yield (GY), plant height (PLH), and 1000-seed weight (TSW) were recorded for all crops (durum wheat, barley, and lentil). Days to heading (DTH) was recorded at stage 55 on Zadok's scale for cereals [58], while for lentil days to 50% of flowering (D50F) was recorded following the lentil ontology [59]. Days to physiological maturity (DTM) was recorded at stage 92 on Zadok's scale for cereals, and from sowing until 90% of the pods are golden brown for lentil. Climatic variables were recorded daily at each site.

The harvested seeds of durum wheat, barley, and lentil trials conducted in Morocco during both seasons were analyzed for major quality characteristics at the Cereal and Legume Quality Laboratory of ICARDA. Grain protein content (GPC) was measured by near-infrared reflectance method (NIR, DS 2500, FOSS, Hillerod, Denmark) using previously validated calibrations. Zinc (Zn) and iron (Fe) concentrations were determined following a modified diacid protocol through inductively coupled plasma-optical emission spectroscopy (ICP-OES); (iCAP-7000 Duo, Thermo Fisher Scientific, Waltham, MA, USA) [60].

In addition, whole grain flour samples of durum wheat obtained with a Udy-Cyclone mill (Retsch GmbH, Haan, Germany) and 0.5 mm sieve was used to determine yellow pigment index (YI) on a chromameter (model CR-5, Konica Minolta Inc, Osaka, Japan). The same flour was analyzed with a mixograph (National Manufacturing Co, Lincoln, NE, USA) according to AACI Method [61] to define the mixograph score (MIXO). Furthermore, β-glucan content was measured in barley seeds via the Skalar San system method (Skalar Analytical BV, Breda, The Netherlands). Measurement of grain length (GL) and grain width (GW) of durum wheat, barley, and lentil was achieved by using the Grainscan method [62].

### 2.4. Data Analysis

In this study, each crop was treated individually for all statistical analysis. Analysis of variance (ANOVA) was done using Genstat program version 21.1 [63]. Multiple comparisons were made with Fisher's least significant difference (LSD) method at a significance level of 0.05. Best linear unbiased estimates (BLUEs) were calculated for various traits across all environments using META-R (Multi Environment Trial Analysis with R for

Windows) version 6.0 [64], considering both genotypes and environments as fixed effects. Broad-sense heritability was calculated based on the following formula:

$$H^2 = \frac{\sigma_g^2}{\sigma_g^2 + \frac{\sigma_{g \times e}^2}{\sigma_E} + \frac{\sigma_e^2}{\sigma_E \times n_r}} \tag{1}$$

where, $\sigma_g^2$ is the genotypic variance, $\sigma_e^2$ is the error variance, $\sigma_{g \times e}^2$ is the G×E interaction variance, $n_r$ is the number of replicates, and $n_E$ is the number of environments in the analysis.

The ratio of variance counted for each source of variations (G, E, and G×E) was calculated dividing the sum of square of each by the total sum of the square for each trait. Only eight accessions of barley were tested in Lebanon, Senegal, and Ethiopia. These were included in the broader analysis by conducting a separate ANOVA using the same models, but with only a subset of genotypes.

For grain yield, G×E was computed into its principal components by additive main effects and the multiplicative interaction 2 (AMMI) model using R software (version 4.0.3) on R studio (version 1.3.1093). The 'AMMI wide adaptation index' (AWAI) was calculated to determine the overall yield stability of each genotype using the following formula:

$$AWAI = \sum_i S_i \, |PC_i| \tag{2}$$

where, i is the number of significant interaction principal component (IPCs) determined by classical Gollob F-test using *agricolae* R package [65], $S_i$ is the percentage of total G×E variance explained by each IPC, and PC (principal component) is the actual IPC value. The AWAI index was derived by summing for each genotype the absolute value of each IPC multiplied by the fraction of the sum of the square explained by each PC. The ratio to maximum value of grain yield (BLUEs) was measured for each environment to remove the effect of irrigation. As described in Bassi and Sanchez-Garcia [66], a biplot between the genetic yield potential (BLUE of grain yield) and yield stability (G×E explained by AWAI) was used to determine the best genotypes combining both G and G×E for grain yield. The AWAI index was presented as ratio to maximum value, and values close to '0' were obtained for the most widely adapted and stable genotypes. However, grain yield values close to '1' indicate the best yielding genotypes across environments. For barley, only the 9 experimental trials conducted in Morocco were included, whereas the 8 selected top yielding lines were analyzed separately, and the ANOVA was performed using Genstat software.

A climate matrix was developed for each environment, with values divided into five growth stages: one month before sowing, sowing to end of vegetative stage, flowering stage, grain filling period, and physiological maturity period. Pearson correlation analysis was conducted between the climatic matrix and the response of genotypes at each site for PLH, DTH (D50F for lentil), and GY. Those climatic factors having a significant effect ($p < 0.05$) were used to perform hierarchical clustering among environments using Ward's method based on Euclidean distance via *dendextend R* package [67].

## 3. Results

### 3.1. Combined Analysis of Variance

For the three crops, the combined analysis of variance (ANOVA) across environments revealed statistically significant differences ($p < 0.001$) for the genotypes (G), environments (E), and their interaction (G×E) for most of the traits (Table S4). The E factor explained the vast majority of the variation for GY, DTH (D50F for lentil), DTM, and PLH in durum wheat, barley, and lentil. In the case of GY, the effect of E was 85.24% in durum wheat, 67.19% in barley, and 72.11% in lentil. Furthermore, the effect of E was the most important source of variation for GL and GW in durum wheat, with 46.98% and 76.58% of the total

variance, respectively. In contrast, the G factor explained the large variation for GL and GW in barley and lentil, as well as for TSW in lentil. The G×E interaction showed a larger contribution to the total variability compared to the G effect for GY, DTM, and PLH in the three crops, and for TSW in both durum wheat and barley. High heritability was observed for TSW, DTH (D50F for lentil), DTM, GL, and GW in durum wheat, barley, and lentil, ranging from 0.61 to 0.98. Furthermore, heritability for GY overall environments was 0.79 in durum wheat and 0.70 in barley, while it measured 0.53 in lentil. The maximum GY was 3981, 3197, and 1113 Kg ha$^{-1}$ for durum wheat, barley, and lentil, respectively, with a coefficient of variation (CV) of 12.92%, 18.56%, and 25.62% (Tables S4 and S5).

In addition, the ANOVA revealed significant variation for E, G, and G×E ($p < 0.001$) among the eight selected genotypes of barley for all the traits across the environments assessed. The overwhelming majority of the variance was explained by the E factor, which explained from 69.41% to 99.76% of the total variance for all variables. For the GY and TSW, the effect of G was restricted to 1.83% and 6.09%, while the G×E effect was 3.73% and 22.78%, respectively (Table S6).

For the quality parameters of durum wheat, the ANOVA showed that the magnitude of the G effect was higher than the magnitude of G×E interaction for MIXO (47.71%) and YI (35.95%), while GPC and Fe content were better explained by the G×E interaction with 34.59% and 39.22% of the total variation, respectively. However, the E factor explained more than 69% of Zn variance, while the effect of G was limited to 2.94% and the G×E effect was 9.90%. The heritability values were 0.65, 0.20, 0.29, 0.79, and 0.93 for GPC, Fe, Zn, YI, and MIXO, respectively (Table S7).

In barley, the E factor explained the largest part of the total variation for GPC, Fe, and Zn, with 81.76%, 54.66%, and 49.98%, respectively. However, β-glucan variation was primarily explained by the G×E interaction (38.09%), and the total variance explained by E was 28.67%. The heritabilities for GPC, Fe, Zn, and β-glucan were 0.72, 0.20, 0.10, and 0.33, respectively (Table S7).

In lentil, the G×E factor explained 55.12% and 46.22% of the total variation for Fe and Zn content, while the E effect reached 23.12% and 32.11%, respectively. However, the large majority of the total variance for GPC was explained by the E factor (54.78%), while G and G×E effects explained 22.92% and 15.09%, respectively. The heritability values of GPC, Fe, and Zn were 0.79, 0.63, and 0.45, respectively (Table S7).

*3.2. Grain Yield Potential and Stability across Environments*

3.2.1. Durum Wheat

A total of 19 environments were used to assess the performances of the CWR-derived lines of durum wheat. The average of GY performance at each station is shown in Table S2. The top-yielding environments were Kfardan20, Marchouch19, Melk Zhar20, and Tessaout19, with Terbol in Lebanon recording top yields of 6365 kg ha$^{-1}$ in 2018–2019 and 6220 kg ha$^{-1}$ in 2019–2020. The lowest GY was measured at the drought affected stations of Sidi el Aidi (both seasons) and Marchouch20 in Morocco, with an average of 1777, 1714, and 858 kg ha$^{-1}$, respectively. Under the extreme heat stress conditions of Fanaye station, the average of GY was measured at 2946 kg ha$^{-1}$. The shortest cycle was recorded in Fanaye, with 55 DTH and 85 DTM, while the longest cycle was recorded in Terbol, with an average of 185 DTM.

The combined analysis of G×E yield stability (AWAI) and G yield potential (BLUE) presented in Figure 2 identified six CWR-derived lines and two checks as superior. These are: Jabal (GID: 800018569), Sahi (GID: 800018532), IDON39-18 (GID: 80001458), Maghrour (GID: 800032178), ADYTM18-099 (GID: 800034117), Nachit (GID: 800043267), Zagharin2, and Kundermiki. Among these, ADYTM18-099 (GID: 800034117) was the top yielder and it includes both *T. dicoccoides* and *Ae. speltoides* in its pedigree.

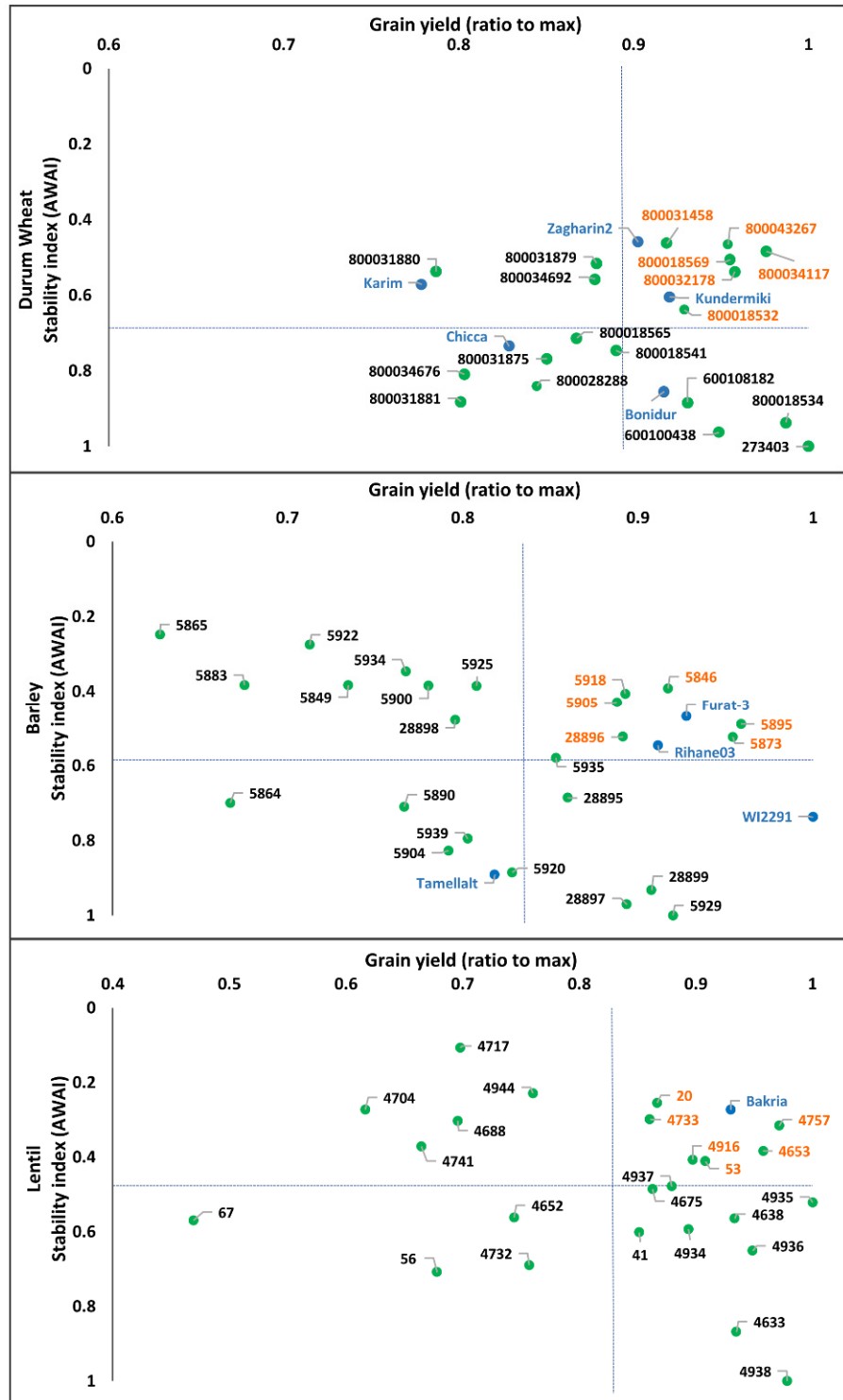

**Figure 2.** Combined analysis of nineteen environments for durum wheat, nine environments for barley, and nine environments for lentil presented as a biplot between genetic yield potential (G) and yield stability (G×E), expressed as the additive main effects multiplicative interaction (AMMI) wide adaptation index (AWAI). All values are presented as ratio to the maximum. Blue color is used to show the checks and green color to mark the CWR-derived lines. The grey lines correspond to the averages. The best yielding and the most stable CWR-derived lines are grouped in the top right corner and color coded in orange. All CWR-derived lines are presented by their GID number.

### 3.2.2. Barley

In total, nine environments in Morocco were used to test all CWR-elites of barley, while only the six CWR-derived top yielders were tested in six additional environments in Ethiopia, Senegal and Lebanon. The highest yields were recorded at Annoceur20, Kfardan20, and Terbol20, with average yields of 4431, 4977 and 5428 kg ha$^{-1}$, respectively, while Debre Zeit20, Fanaye20, and Dheraa20 recorded the lowest yields. Debre Zeit20, Dhera20, and Fanaye20 registered the shortest cycles with less than 60 DTH. In contrast, Terbol20 and Kfardan20 had the longest cycles, with 129 to 135 DTH (Tables S2 and S8).

The combined yield stability (G×E) and yield potential (G) analysis identified six CWR-derived elites (GIDs: 5846, 5895, 5873, 5918, 5905, 28896) and two checks (Furat-3 and Rihane03) as the top (Figure 2). Among these GID: 5895 and GID: 5873 had the highest yields, while the check WI2291 was the overall the top yielder, but was not stable.

### 3.2.3. Lentil

A total of nine environments were used for the evaluation of CWR-derived lines of lentil. The top-yielding environment overall was Terbol20 in Lebanon with an average of 2187 kg ha$^{-1}$. The lowest-yielding stations were Marchouch20, Sidi el Aidi20, Fanaye20 with average yields of 808, 237, and 327 kg ha$^{-1}$, respectively, due to severe drought and temperature stresses (Table S2). In Fanaye20 and Dheraa20, the lentil genotypes experienced the shortest cycle, with just 55 D50F and reached physiological maturity at 95 days. The results of the AWAI study identified six CWR-derived lines (GIDs: 20, 4733, 4916, 53, 4757, and 4653) as the most stable and top yielding genotypes, together with the standard check Bakria. The CWR elite GID: 4935 was the top yielding line across all locations, but was low in stability (Figure 2).

### 3.3. Agro-Climatic Clustering of Test Environments

The durum wheat performances for GY were influenced negatively ($p < 0.05$) by the daily maximum temperature during the vegetative stage [$T_{max(VS)}$] and the average daily maximum temperature during the flowering period [$T_{max(F)}$], and positively by the total water input during the vegetative stage [$WI_{(VS)}$] (Table S9). In the case of barley, GY was positively influenced by the average daily maximum temperature during grain filling [$T_{max(GF)}$] and the water input during grain filling [$WI_{(GF)}$]. For lentil, the average daily maximum temperature during the vegetative stage [$T_{max(VS)}$] had a negative effect on GY, while $WI_{(VS)}$ abundance significantly promoted this trait. In addition, a significant positive impact of $WI_{(VS)}$ was identified on PLH in durum wheat, while both $WI_{(vs)}$ and $WI_{(GF)}$ advanced PLH in barley. However, there was no significant impact of climatic factors on the PLH in lentil genotypes. Furthermore, six covariables ($T_{min(BS)}$, $T_{min(VS)}$, $T_{min(F)}$, $T_{max(BS)}$, $T_{max(VS)}$, and $T_{max(F)}$) showed significant negative effects on DTH in durum wheat, and two ($T_{min(BS)}$ and $T_{min(VS)}$) on D50F in lentil, while no significant impact on DTH was identified in barley.

The significant climatic factors ($T_{max(VS)}$, $T_{max(F)}$, $T_{max(GF)}$, $WI_{(VS)}$, $WI_{(GF)}$) influencing GY were used for hierarchical clustering among environments to define five major agroclimatic groups (Figure 3). The first group (E1) included four environments from Lebanon: Akkar20, Tal Amara19, Kfardan20, and Tel Amara20. The second cluster (E2) included three Lebanese environments: Terbol19, Terbol20, and Kfardan19. The two clusters resulted in high yields thanks to the longer crop cycle and high moisture availability, but were differentiated by higher $T_{max(F)}$ in E1, ranging from 21 to 27 °C, compared to 8 to 20 °C in E2. Fanaye20 in Senegal generated its own cluster (E3) due to the higher maximum temperatures during vegetative growth and flowering time, ranging between 31 and 36 °C. The fourth cluster (E4) incorporated five environments from Morocco: Jemaa Shaim19, Marchouch19, Marchouch20, Sidi el Aidi19, and Sidi el Aidi20. These represent the driest locations with the lowest moisture availability during the vegetative stage. The last cluster (E5) included Moroccan and Ethiopian locations: Annoceur19, Annoceur20, Tessaout19, Tessaout20, Melk Zhar20, Dheraa20, and Debre Zeit20. These had average maximum tem-

peratures during the vegetative and flowering stages, and a moderate moisture availability. This cluster includes mountain stations, Ethiopian medium and lowland sites, and irrigated savannah types (Melk Zehr and Tessaout). Hence, the main evidence for clustering together is a more moderate G×E effect and overall average climatic conditions without extremes.

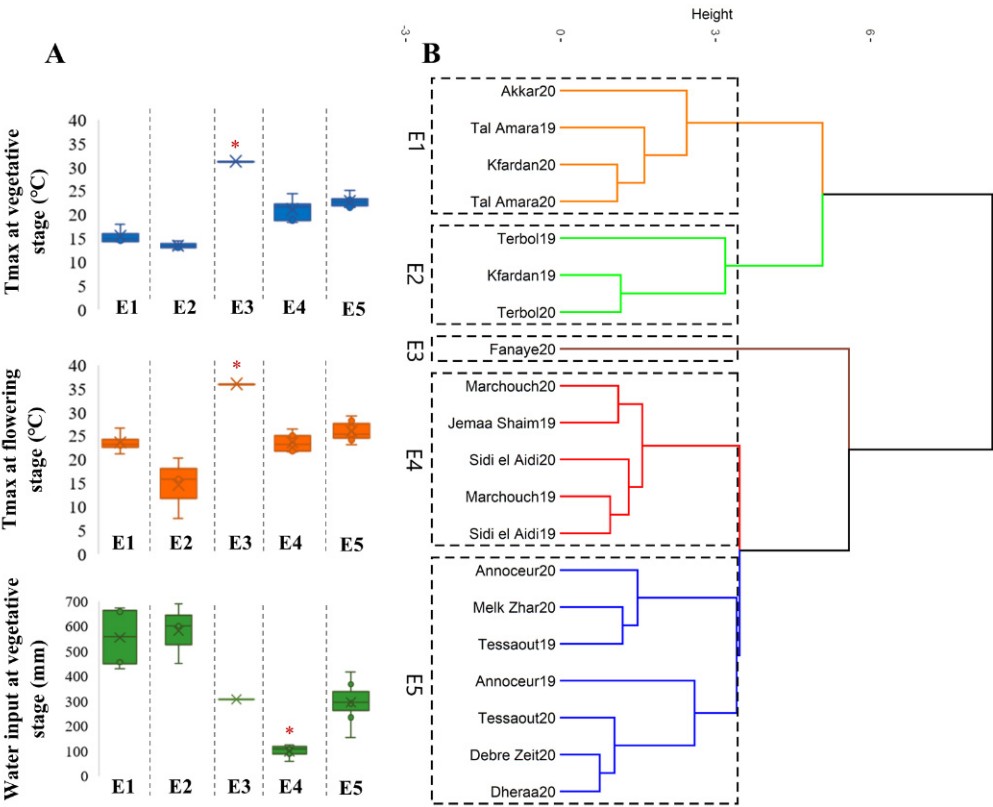

**Figure 3.** Box whiskers of variation for critical climatic factors (**A**) across hierarchical clusters of environments (**B**) determined based on climatic factors having a significant effect on grain yield of durum wheat, barley, and lentil genotypes (maximum temperatures at vegetative stage, maximum temperatures at flowering stage, and water input at vegetative stage). "*" indicates a significantly different response for the trait.

ANOVA among the five clusters showed that the clustering accounted for a significant ($p < 0.001$) fraction of GY variation equivalent to 30.24% in durum wheat, 20.35% in barley, and 43.59% in lentil (Table 1). In addition, the averages of GY were significantly different ($p < 0.05$) among the clusters of each crop. For durum wheat, the lowest average GY (2774 kg ha$^{-1}$) was obtained in E4, E3 registered 2945 kg ha$^{-1}$, while 3401 kg ha$^{-1}$ was the average of GY in the E5. Similarly, E1 and E2 of barley also registered the highest average GY at 4977 and 5427 kg ha$^{-1}$, respectively. The lowest GY was obtained in E3, with 843 kg ha$^{-1}$, whereas E4 and E5 reached 2341 and 3551 kg ha$^{-1}$, respectively. In lentil, E1 was not used for testing, while E2 confirmed the highest average GY with 2187 kg ha$^{-1}$, followed by E5 with 1130 kg ha$^{-1}$. Fanaye20 (E3) and E4 recorded the lowest GY with 367 and 900 kg ha$^{-1}$, respectively.

**Table 1.** Analysis of variance (expressed as percentage) and mean values of grain yield (kg ha$^{-1}$) across identified clusters for durum wheat, barley, and lentil.

|  | Durum Wheat | Barley | Lentil |
|---|---|---|---|
| Clusters (C) | 30.24 * | 20.35 * | 43.59 * |
| Genotypes (G) | 1.75 | 6.55 | 2.22 |
| G x C | 1.48 | 3.17 | 6.86 |
| | Grain yield average | | |
| E1 | 4423b | 4977b | - |
| E2 | 5388a | 5427a | 2187a |
| E3 | 2945d | 843e | 367d |
| E4 | 2774e | 2341d | 900c |
| E5 | 3401c | 3551c | 1130b |

* Indicates significant differences at 0.001 probability level. Means denoted by a different letter indicate significant differences between clusters ($p < 0.05$).

### 3.4. Traits Influencing GY across Mega-Environments

In each cluster, a correlation matrix was developed to understand the relationship between GY, PLH, TSW, and the phenological traits in durum wheat, barley, and lentil (Table S10). For durum wheat, GY was positively correlated with TSW in E1 and E2 ($r = 0.41$ and $r = 0.63$, respectively at $p < 0.01$). In addition, DTH and DTM positively influenced the GY in E1 and E2, whereas they were negatively associated with GY in E4 ($r = -0.46$ and $r = -0.82$, respectively at $p < 0.01$). In E5, GY was also negatively correlated with DTM ($r = -0.43$ at $p < 0.01$). However, no significant correlation was obtained in E3.

In barley, TSW positively influenced the GY in E4 and E5 ($r = 0.56$ and $r = 0.51$, respectively at $p < 0.01$). Similarly, DTH was positively correlated with GY in E1 ($r = 0.52$ at $p < 0.01$), while no significant correlation was identified in E2 and E3 (Table S10).

In lentil, D50F and DTM were negatively correlated with GY in E3 ($r = -0.61$ and $r = -0.55$, respectively at $p < 0.01$). However, TSW was positively correlated with GY in E3 ($r = 0.58$) and E5 ($r = 0.67$). GY was not significantly correlated with any trait in E2 and E4 (Table S10).

### 3.5. Grain Yield Potential of CWR-Derived Lines against Checks

For all crops, the average yield performance of CWR-derived lines against the checks was not significantly different. However, the comparison between the top GY performing CWR-derived elite and the top check of durum wheat revealed significant differences ($p < 0.05$) in E1, E2, and E4, while no significant difference was identified in E3 and E5 even though CWR-derived elites always achieved superior nominal values (Figure 4). In E1, the CWR-derived elite Zeina (GID: 273403) with *T. araraticum* in its pedigree recorded the highest GY, with 4828 kg ha$^{-1}$ compared to the best check Bonidur at 4554 kg ha$^{-1}$ (+6%). In E2, the CWR-derived elite Maghrour (GID: 800032178) incorporating *T. dicoccoides* reached 6488 kg ha$^{-1}$ against the top check Bonidur at 5734 kg ha$^{-1}$ (+13%). In E4, Icaqinzen (GID: 800018534) derived from *T. araraticum* yielded 3035 kg ha$^{-1}$, which was superior to Zagharin2 at 2443 kg ha$^{-1}$ (+24%).

For barley, only in E2 did the best CWR-derived elite reach an average GY significantly superior ($p < 0.05$) to the best check. In E2, the CWR-derived elite GID: 5918 reached 6136 kg ha$^{-1}$ against the best check Chams at 5584 kg ha$^{-1}$ (+10%). Conversely, at all clusters except E5, the CWR-derived elites had a higher nominal yield than the checks, but the design and environmental effects did not allow the finding of statistically significant differences (Figure 4).

For lentil, the nominal yield for CWR-derived elites was superior to the check Bakria in all environments; however, it was only possible in E2 to identify significant differences ($p < 0.05$) among the grain yield performances between the best CWR-derived elite and Bakria. In E2, the CWR-derived elite GID: 4638 reached an average GY of 3049 kg ha$^{-1}$, while the check Bakria had 2382 kg ha$^{-1}$ (+28%) (Figure 4).

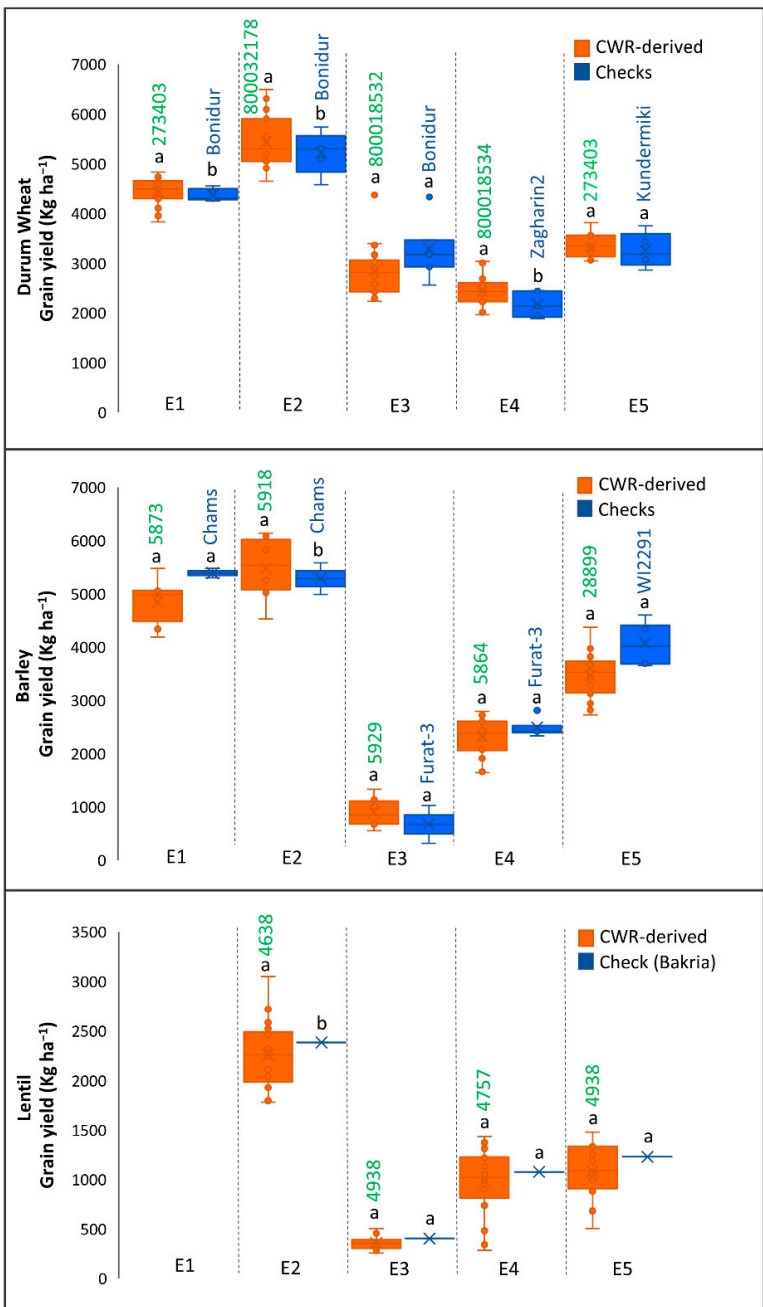

**Figure 4.** Grain yield (kg ha$^{-1}$) of the identified clusters of durum wheat, barley, and lentil. The letters "a" and "b" indicate significant differences using LSD at $p < 0.05$ probability level.

### 3.6. Evaluation of Food Transformation Characteristics

The harvested grains from nine environments for durum wheat, eight environments for barley and eight environments for lentil all located in Morocco were analyzed for the most important food transformation parameters. All the samples of the three crops were analyzed for the following variables: GPC, Fe, Zn, GL, GW, and TSW. Further analysis was performed for durum wheat samples evaluating MIXO and YI, while β-glucan was examined for barley grains.

#### 3.6.1. Durum Wheat

For durum wheat, significant differences ($p < 0.05$) were found between the top CWR-elite and the best check for Zn and Fe content, MIXO score, GL, and GW (Figure 5, Table 2). No significant difference between the top CWR-elite and the best check was identified for

TSW, GPC, and YI, but the CWR-derived elites Nachit and Sahi had the highest nominal value for TSW and GPC combined across environments. Sahi (GID: 800018532) derived from *T. urartu* had the highest value for Zn concentration at 35.84 mg kg$^{-1}$, compared to the best check Bonidur (33.48 mg kg$^{-1}$). For Fe content, the top value was achieved by Icaverve (GID: 600108182), with 30.90 mg kg$^{-1}$ against the best check Karim with 24.70 mg kg$^{-1}$. Finally, the CWR-derived Zeina (GID: 273403) had the top MIXO score at 7.31 compared to the best check Zagharin2 at 5.34. The highest GL was recorded at 7.45 mm by the CWR elite Syryopis (GID: 800018541) derived from *T. aegilopoides* against 7.33 mm of Bonidur, and the highest value for GW was 3.41 mm obtained by Sahi against 3.30 mm by Zagharin2 (Table S11).

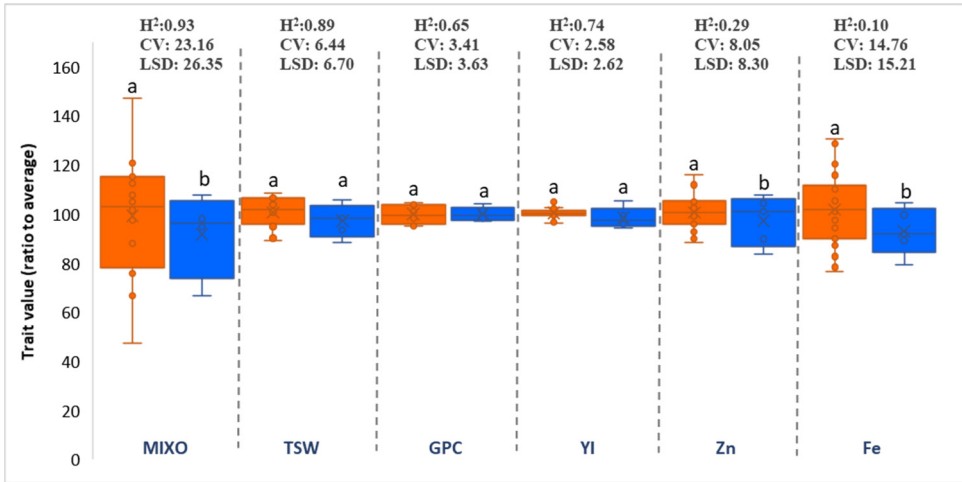

**Figure 5.** Food transformation characteristics of durum wheat CWR-derived elites against checks combined across nine environments in Morocco. The values are plotted as ratio-to-average to be combined in the same graph. Orange color was used to indicate CWR-derived lines and blue color shows the checks. The letters "a" and "b" indicate significant differences at *p* < 0.05. MIXO; mixoscore, TSW; 1000-seed weight, GPC; grain protein content, YI; yellow index, Zn; zinc content, Fe; iron content, H$^2$; heritability, CV; coefficient of variation and LSD; least significant difference.

**Table 2.** Descriptive statistics for grain morphology traits of durum wheat, barley, and lentil across test environments in Morocco.

| Trait | | Durum | | Barley | | Lentil | |
|---|---|---|---|---|---|---|---|
| | | CWR | Checks | CWR | Checks | CWR | Check |
| GL (mm) | Min | 6.76e | 6.98d | 6.22d | 7.67c | 4.41d | - |
| | Max | 7.45a | 7.33b | 8.88a | 8.88a | 6.06a | - |
| | Mean | 7.18c | 7.16c | 8.12b | 8.27b | 5.26c | 5.71b |
| | H$^2$ | | 0.95 | | 0.96 | | 0.98 |
| | LSD | | 0.10 | | 0.27 | | 0.22 |
| GW (mm) | Min | 3.12e | 3.06e | 2.64d | 3.20c | 3.97d | - |
| | Max | 3.41a | 3.30b | 3.41a | 3.37a | 5.52a | - |
| | Mean | 3.25c | 3.18d | 3.27b | 3.31b | 4.80c | 5.22b |
| | H$^2$ | | 0.94 | | 0.97 | | 0.98 |
| | LSD | | 0.06 | | 0.07 | | 0.20 |
| TSW (g) | Min | 37.60c | 37.16c | 40.52d | 42.59d | 24.38d | - |
| | Max | 45.76a | 44.51a | 51.97a | 48.76b | 52.13a | - |
| | Mean | 42.38b | 40.99b | 46.48c | 45.44c | 39.49c | 46.41b |
| | H$^2$ | | 0.93 | | 0.88 | | 0.97 |
| | LSD | | 1.98 | | 2.77 | | 5.85 |

Means for each variable followed by the same letters are not significantly different (*p* < 0.05). GL; grain length, GW; grain width, TSW; 1000-seeds weight.

### 3.6.2. Barley

Among the tested barley germplasm, significant differences were found between the top CWR elite and the best check for all food transformation parameters, except Fe content, GL, and GW (Figure 6 and Table 2). In the case of GL, a maximum value of 8.88 mm was reached by both the best CWR elite GID: 5875 and the check Furat-3, while the CWR line GID: 5918 was the top for GW with 3.41 mm against the check WI2291 with 3.37 mm. The CWR-derived line GID: 5883 registered the highest GPC at 13.96%, while the check Tamellalt reached 12.53%. For TSW, the highest value was recorded by the CWR-derived line GID: 5929 at 51.97 g, whereas Furat-3 was the top check with 48.76 g. The CWR-derived elite GID: 28,898 had the highest Zn concentration with 37.38 mg kg$^{-1}$ against 34.89 mg kg$^{-1}$ of the check Tamellalt. The check Rihane03 accumulated the highest Fe concentration with 41.52 mg kg$^{-1}$, followed by the CWR-derived lines GID: 28,898 (41.33 mg kg$^{-1}$). The highest value for β-glucan was reached by the CWR-derived elite GID: 5925 at 5.81%, against 5.07% of Rihane03 (Table S11).

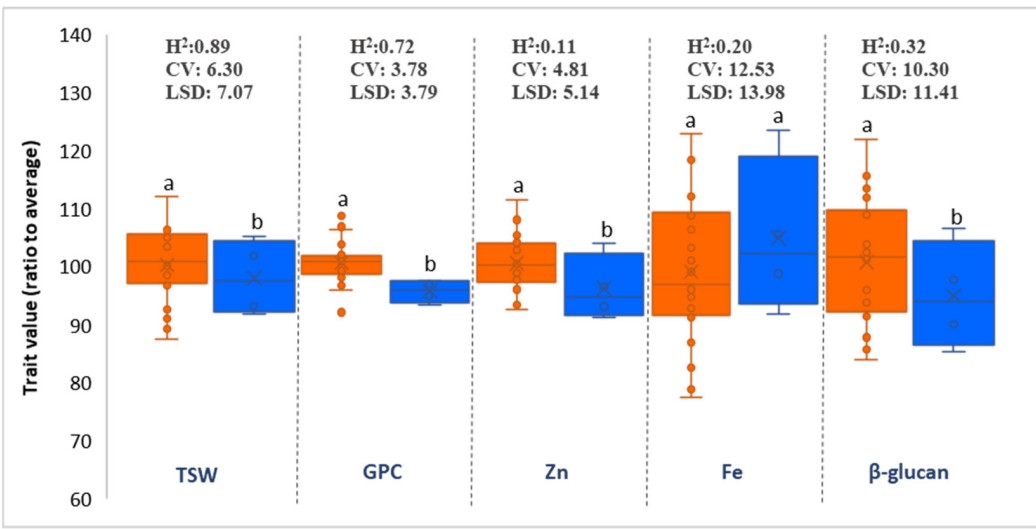

**Figure 6.** Food transformation characteristics of CWR-derived elite lines of barley (orange) against commercial checks (blue). Values are expressed as ratio-to-average to plot them together. The letters "a" and "b" indicate significant differences at $p < 0.05$. TSW; 1000-seed weight, GPC; grain protein content, Zn; zinc content, Fe; iron content.

### 3.6.3. Lentil

For lentil, significant differences at $p < 0.05$ was identified between the top CWR-derived elite and the check Bakria for Zn and Fe content, GL, and GW, while no significant difference was obtained for TSW and GPC (Figure 7 and Table 2). Conversely, the CWR-derived elite GID: 20 had the highest TSW (52.13 g) and GPC (24.27%), compared to Bakria, which had 46.41 g and 23.48%, respectively, even though these are only nominal, non-significantly different values. Instead, for Zn content, the highest value was achieved by the CWR GID: 41 at 61.24 mg kg$^{-1}$ against 52.10 mg kg$^{-1}$ of the check. Further, the CWR GID: 4652 accumulated the highest Fe concentration with 78.24 mg kg$^{-1}$, while Bakria recorded only 52.11 mg kg$^{-1}$. The CWR GID: 20 had the highest value for both GL (6.06 mm) and GW (5.52 mm) across environments, exceeding the check Bakria at 5.71 mm and 5.22 mm, respectively (Table S11).

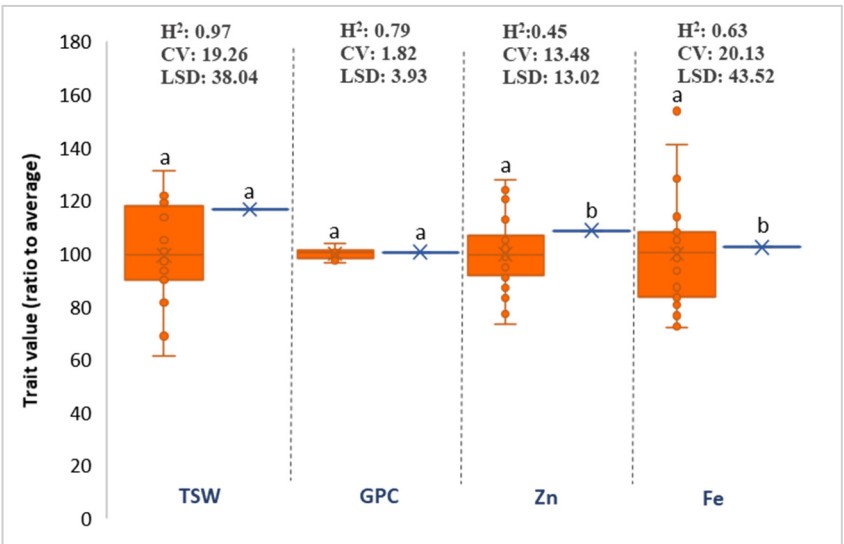

**Figure 7.** Food transformation characteristics of CWR-derived elite lines of lentil (orange) against the commercial check Bakria (blue). Values are expressed as ratio-to-average to plot them together. The letters "a" and "b" indicate significant differences at *p* < 0.05. TSW; 1000-seed weight, GPC; grain protein content, Zn; zinc content, Fe; iron content.

## 4. Discussion

### 4.1. CWR-Derived Elites Yield Performances Combined across Sites

Elites of durum wheat, barley, and lentil derived from CWR were assessed for yield performances against top commercial checks across 19 environments located in Morocco, Ethiopia, Senegal, and Lebanon. Due to the vast geographical scope of the current study, it is not surprising that the E factor accounted for the vast majority of the variation, ranging from 67.2% for barley to 85.2% for durum wheat. Similar results have been reported frequently in the literature [66,68–71] when a high number of diverse environments was used for testing. Here, the research farms varied from the heat-affected shores of the Senegal River, to the very dry conditions of several regions of Morocco and Ethiopia, to various high input environments in Lebanon and Morocco, all the way to the frost-affected sites in the Atlas Mountains. The G×E interaction had an expectedly strong impact on GY (*p* < 0.001), even though broad-sense heritability was high for all crops, from $H^2 = 0.53$ in lentil to $H^2 = 0.79$ in durum wheat.

Because of the relatively high G and G×E impact on GY, it is of interest to compare the overall yield potential (G) and yield stability (G×E) of the CWR-derived elites against the checks (Figure 2). Environment and genotype x environment interactions were reported as the important components for the assessment of genotype adaptation and for controlling the expression of GY [72,73]. Therefore, selection should be attained in various environments, and different genotypes should be selected for specific environments to identify the best stable and high yielding genotypes [74]. For that, stability analysis using the AMMI wide adaptation index (AWAI) was adopted. The genetic yield potential and yield stability were used to determine the best genotypes combining both G and G×E for grain yield. The results revealed six CWR-derived elites and two checks of durum wheat resulted among the most promising, with ADYTM18-099 (GID: 800034117) incorporating both *T. dicoccoides* and *Ae. speltoides* in its pedigree being the nominal top yielder. The best performing CWR elites of durum wheat were derived by top crosses using *Ae. speltoides*, *T. araraticum*, *T. dicoccoides* and *T. urartu*. These results are supported by several investigations that described the usefulness of using these progenitors in breeding durum elites with high yield and resistance to major biotic and abiotic stresses [8,25,47,74,75].

Among the barley germplasm, six CWR-derived elites and two checks were also the superior lines across environments, with two elites derived from *H. spontaneum* (GID: 5895 and GID: 5873) being the top yielders. These findings demonstrate the potential

of *H. spontaneum* to enhance barley yield potential, which is in strong agreement with the observations of Lakew et al. [52] and Bahrami et al. [76]. Likewise, among the lentil genotypes, six elites derived from *L. orientalis* and the check Bakria had the best combined performances. This supports the outcomes of Idrissi et al. [77] in which Bakria showed wide adaptation and stability across 14 different environments, including drought and heat stress sites. However, our results showed that three new CWR-derived lines exceeded the GY of Bakria, with the CWR elite GID: 4935 being top nominal yielder across environments.

Therefore, for all three crops, the combined analysis revealed the clear superiority of the CWR-derived elites compared to the checks. This is in alignment with other studies reporting the value that CWR-derived alleles bring to the performances of elite lines [2,78–83]. To the best of our knowledge, this is the first study in which three crops are studied jointly for the contribution that CWR relatives have to improving the yield performances of elite germplasm. Because of the vast geographical scale of our study and its multiple-crop nature, it seems adequate to conclude that the inclusion of CWR in the pedigree did not negatively affect the yield performances and, on the contrary, performances above the checks could be obtained, even if the study involved only a limited number of entries.

### 4.2. CWR-Derived Elites Yield Performances across Clusters of Environments

The overall G and G×E performances are good indicators of the potential for genetic advancements using CWR. However, the E factor accounted for the largest part of the variation. To reduce the E effect and achieve better targeting of varieties, breeders tend to select germplasm well adapted to one targeted population of environments, rather than entries with average good performances across diverse sites. For this scope, the 19 environments were grouped into five clusters based on their recorded climatic conditions at critical developmental stages. In particular, maximum temperature at the vegetative stage and floral development stage, and moisture availability during the vegetative stage were identified as the main climatic factors affecting GY and, hence, were used for clustering (Figure 3). This clustering procedure captured good portions of the E variation for GY, ranging from 20.35% in barley to 43.59% in lentil. Recently, Ayed et al. [74] also reported similar outcomes in which water input and temperature appeared to be the major climatic factors influencing the response of durum wheat genotypes, however, Ndiaye et al. [84] suggested that soil fertility, water input, and sowing date were the critical factors of sorghum genotypes.

In our study, the first cluster (E1) included sites in Lebanon with moderate maximum temperature and high moisture content during the vegetative stage. Lentil was not tested in this cluster. For barley, the performances of CWR-derived and checks was not significantly different; however, the CWR-elite GID: 273,403 exceeded all the commercial checks over environments and generated the highest grain yield. For durum wheat, the CWR-derived elite Zeina (GID: 273403) significantly outyielded the best check by 6%. The second cluster (E2) represents the most productive sites in Lebanon, reaching the highest average yields thanks to the lowest maximum temperatures before flowering and highest moisture availability. Here, for all three crops, it was possible to identify CWR-derived elites significantly superior to the checks, such as Maghrour derived from *T. dicoccoides* for durum wheat (+13% yield gain), GID: 5918 derived from *H. spontaneum* for barley (+10% yield gain), and GID: 4638 derived from *L. orientalis* for lentil (+28% yield gain). Cluster E3 was represented by one station in Senegal in which maximum temperatures at vegetative and flowering times ranged between 31 and 36 °C. Here, at least one CWR-derived elite for each crop outperformed the best check, but not significantly. This is in partial disagreement to what was reported by Sall et al. [8] and El Haddad et al. [47] for the same site, who had identified germplasm derived from *A. speltoides* as the top performing one for durum wheat, thanks to its ability to maintain fertile pollen despite the heat stress. Probably the use of one single season of testing and the inclusion of even better performing checks prevented this study from reaching the same results. Cluster E4 incorporated sites from Morocco, which received the lowest amount of moisture during the vegetative stage. For durum

wheat only, the elite Icaqinzen derived from *T. araraticum* obtained a significantly superior yield compared to the best check, with a 24% yield advantage. For barley and lentil, the CWR-derived lines generated the greatest yield in E4, exceeding the commercial checks, but not significantly, and this indicates the high performance of the checks used across environments. The high adaptation of the checks in E4 with only Moroccan sites could be explained by the fact that they were widely used as the best control lines in the identification of genotypes with favorable characters. Finally, cluster E5 grouped together Moroccan and Ethiopian environments, ranging from mountain stations to Ethiopian medium and lowland sites, and irrigated savannah types (Melk Zehr and Tessaout). Hence, no clear climatic trends grouped these sites, except a tendency toward an absence of extremely high temperatures and drought stress. As for E3, no significant yield superiority could be identified for any of the three crops, with check results matching those of CWR-derived entries.

The conductive effect of E2, where CWR-derived elites superior to the checks could be identified for all crops, seems to suggest that CWR alleles are most advantageous under high yield potential conditions. This is somewhat in disagreement with the literature that instead tends to identify a stronger potential contribution of CWR to adapt elite germplasm to stresses, rather than for reaching higher yield potential [49,85–87]. Nevertheless, the nominal yield at most clusters also shows a positive contribution of CWR in stressed environments. Hence, it might be possible that the higher experimental error typical of drought or heat stressed environments is somewhat masking the significance of the CWR contribution. In general, it appears clear that the use of CWR in the crossing schemes of durum wheat, barley, or lentil is not causing any detrimental effect on yield. Rather, it is raising the overall yield potential and yield stability, and the adaptation to high input environments. Improving yield performance using CWR has been reported in several crops such as durum wheat [8,25,47,74], barley [52,88], maize [78], lentil [87,89,90], and chickpea [91,92]. Hence, it can be suggested that breeders should look at CWR as a valid source of yield increasing alleles, rather than as a possible cause of linkage drag.

Correlation coefficient analysis was conducted to determine the traits that were most closely associated with the variation of grain yield for each cluster and crop. In durum wheat, the results showed high significant correlation between GY and TSW in the two mega-environments E1 and E2. Therefore, TSW appears as one of the most important yield components that significantly contributed to the variation of genotypes for GY, which can explain the high performance of CWR-derived lines over the checks in E1 and E2. These outcomes support our previous findings showing that CWR-derived lines of durum wheat had larger grain size over cultivars and elites [47]. In addition, DTH and DTM were positively correlated to GY in both clusters of Lebanon, which suggests that phenology also represents a critical characteristic for yield formation. Several investigations reported that the extension of the grain filling period and water availability for photosynthesis are the major factors to reach high yields in wetter and cooler environments of the Mediterranean Basin [93–96]. In contrast, DTH and DTM were not significantly correlated with GY under the heat stress of E3. Similar results were obtained by Sall et al. [8], who also concluded that phenological traits did not affect the yield of durum wheat genotypes under the high temperatures of the Fanaye station, and indicated no significant difference between top and bottom-yielding lines for phenology. In E4, which comprised only drought-prone Moroccan sites, our results revealed that phenological traits were highly negatively correlated with GY. This correlation might highlight the importance of earliness in escaping more drought stress in later stages, in addition to promoting an efficient use of available resources to maintain high yield potentials. Aberkane et al. [82] suggested that the selection of early genotypes with long grain filling period could lead to selecting lines with good levels of drought tolerance.

In barley, the TSW showed a positive correlation to GY in E4 and E5, which could explain the superiority of top CWR-derived lines over the best check. These results are in good agreement with the findings of Carpici and Celik [97], who reported a direct effect

of TSW on grain yield in barley genotypes. In addition, DTH had a positive correlation with GY in E1, while the impact of DTM was not significant. This would suggest that the heading date plays more decisive role in the determination of yield potential in E1, rather maturity period. In lentil, TSW was the most positive direct contributor towards GY in E3 and E5, and could be relied upon for the selection of genotypes to improve the genetic yield potential of lentil under extreme heat stress, as well as in the favorable conditions of Morocco and Ethiopia. However, no significant impact of TSW on GY was found in E2 and E4. In addition, it was observed that D50F and DTM had significant negative association with GY in Fanaye (E3), which is in agreement with previous studies that reported similar findings and highlighted the importance of early flowering and maturity in response to heat stress in lentil [98,99].

*4.3. Better Food Transformation Characteristics in CWR-Derived Elites*

The harvested grains of durum, barley, and lentil are used by the transformation value chain to produce various types of foods. It is therefore essential that beyond better adaptation and higher productivity, the use of CWR does not negatively affect the suitability of the harvests to be used by the food value chain. The E and G×E factors provided the most significant amount of variation for all end-use quality traits of the three crops, with the exception of YI and MIXO for durum wheat, where the effect of G was dominant. This is in agreement with the findings of Zaim et al. [25] and Boehm et al. [100], who identified strong G control for YI and MIXO in durum wheat. Moreover, broad sense heritability was defined as moderate to high for most of the measured traits, indicating that a certain degree of G control could be identified for all traits. Zn and Fe concentration in durum wheat and barley had low heritability, making the genetic improvement for these traits extremely challenging. For these traits, it is evident from our study that the use of different soil types and farming conditions have a stronger effect than the genetic predisposition for accumulating the nutrients. This is in partial disagreement with previous studies [101–103], in which the identification of cereal elites with higher genetic propensity for Zn and Fe accumulation in the grains was possible. Our results tend to suggest that in grain legumes there is stronger genetic control for micronutrient content, making the identification of nutrient-rich lentil elites more meaningful.

For durum wheat, our results indicated no significant difference between the top CWR-derived line and the best check for TSW, GPC, and YI. Hence, the addition of CWR in the pedigree did not negatively affect the overall end-use quality and transformation characteristics of the germplasm. This is in partial disagreement with what was reported by Zaim et al. [25], El Haddad et al. [47], Farooq and Siddique [104], and Mondal et al. [105], who suggested that CWR can cause a linkage drag in terms of end-use characteristics in durum wheat. However, these studies had also identified a positive contribution of CWR (especially *T. dicoccoides*) for the improvement of grain size (TSW). Therefore, the remarked difference in results might be due to the much larger geographical scope and depth of analysis conducted here. The current study highlighted the ability of several CWR-derived elites to outperform the best check for the MIXO score, Zn, and Fe. Still, even if statistically superior elites derived from CWR were identified for their content of these two micronutrients, the actual values are too small to justify seeking their commercial use. More interestingly, the CWR-derived elite Zeina had a MIXO score across locations averaging 7.31, which is indicative of outstanding pasta and bread making quality. This result, combined with Zeina's top yield performances recorded for E1 and E5, point to a positive contribution of *T. araraticum* to the overall performances of the durum elite germplasm.

For barley, our results showed high and significant superiority of the top CWR line compared to the best check for all quality traits, excluding Fe content. Especially for GPC, β-glucan content, and TSW, the contribution of *H. spontaneum* was extremely evident, with several elites outperforming the best checks. This is in good agreement with the findings of Hebelstrup [54], who also reported a clear advantage of *H. spontaneum* for these

traits compared to domesticated barley. In addition, the CWR-derived elites superiority for Zn and Fe content identified here has also been observed in diverse studies [51–53]. Remarkably, the CWR-derived line GID: 5918 combined top yielding performance for E2 and high-quality characteristics, especially for GPC and TSW, over the commercial checks.

For lentil, several elites derived from *L. orientalis* had significant higher Ze and Fe concentrations compared to the check Bakria. Similar results were observed by Kumar et al. [1] in which *L. orientaltis* was among the best source for high Zn and Fe. In this study, the broad sense heritability for micronutrient concentration in lentil trials reached values two to four fold superior to durum wheat and barley, suggesting that the genetic biofortification of grain legumes is more easily achievable than in cereals. However, lentil CWR-derived elites matched the check in terms of grain weight and protein content, which is in partial disagreement with the findings of Khazaei et al. [56] and Abbo et al. [106]. These authors had in fact reported that *L. orientalis* could promote the doubling of the protein content in the grains and seed weight, but our in-depth study was unable to confirm this result.

## 5. Conclusions

In our study, it was possible to confirm that for three critical dryland crops (durum wheat, barley and lentil) the use of CWR in the pedigree did not negatively affect yield performance, climate adaptation, or end-use quality. In contrast, several CWR-derived elites were superior to the checks for yield potential (G) and yield-stability (G×E) combined across environments. When screening for clusters of environments representing five target population of environments, it was possible again to identify CWR-derived elites superior to or matching the top check. Hence, CWR-derived alleles improved not only the yield response to specific climatic conditions, but also across all environments. Finally, when screening for end-use quality, it was not possible to identify any evident linkage drag compared to the check. Instead, for durum wheat, better pasta firmness and bread making quality could be derived from *T. araraticum*, while for barley, very high fiber content (β-glucans) was obtained from *H. spontaneum*, and in lentil, higher concentrations of micronutrients were confirmed in elites derived from *L. orientalis*.

Therefore, our study represents one of the most complete assessments to date of the use of CWR for the genetic improvement of durum wheat, barley, and lentil. Our results strongly support the use of CWR as ideal parental material for the production of top performing elites, without any linkage drag. As such, it was not possible for us to confirm the suggestions by many classical breeding books that the use of CWR as parents had an obvious detrimental effect on the overall performance, and hence, that these should be used only as a last resort. Rather, our data reveal that strong genetic gain for many traits is possible when incorporating CWR, even if a limited number of elites was used. To justify this fact, some authors reaching similar conclusions to ours have been suggesting a recurring hypothesis that the increase in allelic diversity obtained when using CWR crosses can supplement the lower selection intensity due to the lower number of crosses screened compared to elite x elite strategies. Our results do not negate this hypothesis. Still, more detailed molecular analysis would be needed to fully test this hypothesis.

**Supplementary Materials:** The following are available online at https://www.mdpi.com/article/10.3390/agronomy11112283/s1. Table S1: Details of durum wheat, barley and lentil genotypes evaluated during seasons 2018–2019 and 2019–2020, Table S2: Experimental farms used for field evaluation during the seasons 2018–2019 and 2019–2020, Table S3: Essential management practices for durum wheat, barley and lentil on the different experimental sites, Table S4: Analysis of variance for all traits of durum wheat, barley and lentil across locations, the value of each Source of variation (SOV: E, G, and G×E) is presented as ratio of the total variation, Table S5: Grain yield (kg ha$^{-1}$), plant height (cm), and phenological traits for durum wheat, barley and lentil across environments, Table S6: Analysis of variance for the traits of the 8 selected entries of barley across Lebanon, Senegal and Ethiopia environments conducted during 2019–2020 cropping season. the value of each Source of variation (SOV: G, E, and G×E) is presented as ratio of the total variation, Table S7: Analysis of

variance for quality traits of durum wheat, barley and lentil across locations, the value of each Source of variation (SOV: E, G, and G×E) is presented as ratio of the total variation, Table S8: Grain yield (kg ha$^{-1}$), plant height (cm), 1000-seed weight (g) and phenological traits of the eight CWR elites of barley at Terbol20, Kfardan20, Debre Zeit20, Dheraa20 and Fanaye20, Table S9: Pearson correlation between grain yield, plant height, days to flowering and climatic matrix of durum wheat, barley and lentil across all environments, Table S10: Pearson correlation coefficient between grain yield and plant height, days to heading, days to 50%flowering for lentil, days to maturity and 1000-seeds weight per cluster in durum wheat, barley and lentil, Table S11: Average values (BLUEs) for food transformation characteristics of the CWR-derived elites and checks of durum wheat, barley and lentil across environments in Morocco.

**Author Contributions:** Conceptualization and methodology, N.E.H., M.S.-G., A.V., S.K. and F.M.B.; Statistical analysis, N.E.H.; validation, F.M.B.; investigation, N.E.H., R.E.A., A.T.S., W.L. and A.J.; data curation, N.E.H.; writing—original draft preparation, N.E.H.; writing—review and editing, M.S.-G., A.V, S.K. and F.M.B.; project administration, F.M.B. All authors have read and agreed to the published version of the manuscript.

**Funding:** This research was fully funded by Adapting Agriculture to Climate Change: Collecting, Protecting and Preparing Crop Wild Relatives, supported by the Government of Norway, managed by the Global Crop Diversity Trust with the Millennium Seed Bank of the Royal Botanic Gardens, Kew project GS18009: "DIIVA-PR: dissemination of interspecific ICARDA cultivars and elites through participatory research".

**Institutional Review Board Statement:** Not applicable.

**Informed Consent Statement:** Not applicable.

**Data Availability Statement:** Available upon reasonable request.

**Acknowledgments:** The authors wish to recognize the intense work of the field technical staff in Morocco, Lebanon, Ethiopia, and Senegal.

**Conflicts of Interest:** The authors declare no conflict of interest.

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
