# Peer review of "Crop Wild Relatives Crosses: Multi-Location Assessment in Durum Wheat, Barley, and Lentil"

_agronomy, doi:10.3390/agronomy11112283_

Round 1
Reviewer 1 Report
In the present manuscript the authors have analyzed CWR-derived elites of the three crops (durum wheat, barley, and lentil). The chosen elites have been compared individually against the most meaningful checks across in countries where the test fields were in locations with clearly of different climate conditions. The research was concentrated on a detailed assessment of the usefulness CWR (crop wild relatives) in the breeding both for adaptation to climate stresses and for addition of nutritional value to commercial wheat, barley and lentil varieties.
The germplasm was tested in 13 research stations located in Morocco, Lebanon, Ethiopia and Senegal , over two successive crop seasons 2018-2019 (19) and 2019-2020 (20), for a total of 19 environments available for the test.
The manuscript was well written and quite easy to follow. The data was presented in logic order and the figures and tables contained all the necessary data. But I still have some comments and questions as few things remained unclear for me.
Quetsions and remarks
- Table S1. – In this table there is a lot of CWR-derived elites. How were this lines obtained, is this a part of the present work or the authors used the germplasms from some kind of collection? I did not find any reference. How was determined the HP rate of CWR genomic contribution?
- Table S1 – Durum wheat germplasm Sahi – CWR is marked as urartu. Line 411 it says that CWR for Sahi is T. araraticum!?
- 3. Line 114. – As I am do not know very well the geography of the countries where the experiments were done just one question to get some information about it. Are the study sites for field experiments chosen in the main wheat, barley growing areas?
- Line 146. – Do the authors think that two replicates for each experiment is sufficient? I think that usually three replicates are done.
- 8. line 322. Here you refer to figure 3 where presented dada for agro-climating clusters from E1 to E5. But later on page 10, line 360 you refer to Table S8 where the clusters E1 to E5 are named as GY clusters. Is this a mistake or if not please explain!
- Figure 5, 6, 7 – in these figures you present the food transformation characteristics of CWR-derived elite lines as trait value (ratio to average). In the text you describe the GPC, TSW etc values for the best CWR-derived elites as mg kg-1 . Can you somehow show it also on the figure, it’s then easier to follow the situation.
- 13. Line 433. While looking on the results in Table 2 to my eye the mean values between CWR and checks for barley GL, GW and TSW are quite similar (?).
- I was thinking that maybe you might include to the manuscript a figure showing the connection between yield potential between yield potential and food transformation characteristics of CWT-derived barley, wheat and lentil lines. To have wheat or barley line with good yield and good food transformation characteristics will be a dream for the farmers.
Reviewer 2 Report
Dear Authors,
I recognize your intense and useful work.
I enjoyed it.
Here is my few suggestions.
Sncerely
29, 30, 31, 32 lines – name species format is better in italic.
144 line – Are Barley locations 13 or 14? In table S2 are 14 locations.
181-83,187 lines – variance symbol is σ² not δ2.
191- 197 lines – it is not clearly described the AWAI index. It would be better to explain what are PC (principal component) and IPC (interaction principal component) even if reported reference "Bassi and Sanchez-Garcia [67]".
231 line – “The maximum GY was 3981, 3197 and 1113 Kg ha-1 ….” : how are these values calculated?
317 line – “Furthermore, five covariables…”: they are six covariables.
410-419 lines – There is no supplementary table showing CWR-derived elites and checks nominal values of all recorded traits.
421 line – in Figure 5, the word “derived” is repeated twice.
108, 434 lines – Is check name Tamellalt or Tamellat ?
433-440 lines – There is no supplementary table showing CWR-derived elites and checks nominal values of all recorded traits.
450-457 lines – There is no supplementary table showing CWR-derived elites and Bakria checks nominal values of all recorded traits.
655 line – “…many elites outperforming the best checks…..”: change in many CWR-derived elites.
In supplemental tables:
Table S1 – in CWR column, species name format change in italic. Is check name Tamellalt or Tamellat ?
Table S7 – “Tmin, average minimum daily temperature; Tmax, average maximum daily temperature, Rh, relative humidity; WI, water input in mm. In each climatic factor, BS corresponds to one month before sowing; VS, vegetative stage; F, flowering period; GF, grain filling period; M, maturity period. * p<0.05, ** p<0.01.”
Reviewer 3 Report
Comments:
- A better worded and shorter title may be more appropriate. The current one is too lengthy and wordy.
- The text needs language improvement at a number of points.
- The overall presentation needs to be squeezed and made well balanced. Currently it is wordy and lengthy.
Reviewer 4 Report
The work is interesting and its subject matter is very topical in the face of advancing climate change. The introduction fits well with the key context discussed in this manuscript and justifies the purposefulness of the research undertaken. The goals are clear and scientifically sound. It should be emphasized that extensive research material covers the 3 most important arable crops in Africa (durum wheat, barley, lentils), which are an important source of human and animal nutrition, and the geographical scale of the research. The field research was conducted for 2 years in 19 centers located in 4 countries, which allowed for the differentiation of soil and climatic conditions. The results are presented in a very clear graphical manner. The obtained results have a practical aspect.
I propose a few minor changes and fixes listed below:
In the methodology, please complete the data on the manufacturer and country of origin of the devices
used in research
Please complete the units of measurement in tables 1 and 2
In Figure 4, please change the entry of the units (it should be kg ha-1)
Ln - please provide information on the highest yields
Ln 500 - please convert (2019) to [78]
Ln 604 - please remove the redundant period
Ln 952 - please change the case of the spelling of the last name
Reviewer 5 Report
Haddad et al. present a carefully drafted study which conducted comprehensive field trials of three crops in multiple countries and locations. This study confirmed that the use of CWR in breeding did not cause linkage drag to yield related traits, instead some CWR-derived lines improved yield performance and end-use quality characteristics. This study demonstrated the usefulness of CWR to expand the genetic diversity to improve crop yield and quality traits in various climate conditions. I think the manuscript is generally written to a high standard, yet I have a few small suggestions that the authors might find useful.
- Discussion section is a bit long and redundant in some case. If it can be a bit more concise, the readers will appreciate it.
- Minor edits required:
- Remove some extra space/blank
- A few minor grammar error
